# Recent Advances in the Histopathology, Molecular Biology, and Treatment of Kaposi Sarcoma: A Contemporary Review

**DOI:** 10.3390/ijms262010058

**Published:** 2025-10-16

**Authors:** Tayarv Jayd Bagratee, Veron Ramsuran, Mpumelelo Msimang, Pratistadevi Kanaye Ramdial

**Affiliations:** 1School of Laboratory Medicine and Medical Sciences, College of Health Science, University of KwaZulu-Natal, Durban 4041, South Africa; 2Centre for the AIDS Programme of Research in South Africa (CAPRISA), University of KwaZulu-Natal, Durban 4013, South Africa; 3International Foundation for Integrative Medical Research, Cape Town 8001, South Africa; 4Department of Anatomical Pathology, National Health Laboratory Service and School of Laboratory Medicine and Medical Sciences, University of KwaZulu-Natal, Durban 4091, South Africa; 5Department of Anatomical Pathology, Nexomic Laboratories, Durban 4001, South Africa; 6Department of Anatomical Pathology, Walter Sisulu University, Mthatha 5099, South Africa

**Keywords:** Kaposi sarcoma, HHV8, KSHV, clonality, histopathology, molecular

## Abstract

Kaposi sarcoma (KS) is an intermediate-grade vascular tumour that has undergone major treatment and diagnostic breakthroughs following the discovery of Human herpesvirus 8 (HHV8). Whilst classically described in Eastern European populations, the endemic and epidemic forms of KS have facilitated its association with AIDS. This was led by the detection of HHV8 by PCR, and thereafter, immunohistochemically. This not only enabled the recognition and diagnosis of complex histopathological KS subtypes but also facilitated distinction from its mimickers, including acroangiodermatitis and pyogenic granuloma. Recent advances in the viral genomics of HHV8 have expanded the diagnostic landscape of KS clinically and molecularly. The latent phase of replication in the HHV8 lifecycle reveals numerous angiogenic and inflammatory factors. Novel therapies targeting these viral–human molecular interactions may prove useful. However, this is highly dependent on the clonal nature of KS. Conflicting research outcomes demonstrate varying viewpoints on the clonal (monoclonal/oligoclonal/polyclonal) nature of KS, heightening the tumoural versus inflammatory pseudoneoplastic controversy. Understanding the clinical context of KS is fundamental to understanding its clonality, and a dearth of this clinical information in recent studies appears to be the critical factor in determining the true clonal nature of KS. The current molecular landscape, histopathology, treatment options, and opinions on clonality are critically reviewed.

## 1. Introduction

Kaposi sarcoma (KS), recognised since 1872, gained prominence in the human immunodeficiency virus (HIV) infection and AIDS (HIV/AIDS) era and remains the most common worldwide malignancy in people living with HIV/AIDS. The increased KS prevalence not only improved the global understanding of KS biology and outcomes but also enabled the diagnosis of an expanded spectrum of KS phenotypes and mimickers. Whilst this has been underpinned by a heightened awareness of KS and its mimickers, the discovery of Kaposi sarcoma-associated herpesvirus (KSHV), also known as Human herpesvirus 8 (HHV8), as the aetiopathogenetic cause of KS, along with improved molecular and histopathological techniques, have collectively enhanced the biological and diagnostic landscape of KS (Figure 1). In the histopathological diagnostic terrain, the discovery of HHV8 immunohistochemical antibodies, suitable for use on formalin-fixed paraffin wax-embedded tissue, has singularly:Expanded the diagnostic histopathological KS subtypes;Enabled KS diagnosis when it has mimicked inflammatory, infective, and pseudotumoural conditions;Distinguished KS from other soft tissue spindle cell sarcomas.

The identification of HHV8 as the causative agent of KS is not only one of the most promising discoveries in the KS sphere, but it has also promoted intensified scientific investigations of HHV8 genomic variation in KS development.

The manifold aims of this review are to

Present the past and current histopathological landscape of KS, with a critical analysis of the potential difficulties associated with the diagnosis of KS;Assess the pivotal role of HHV8 in KS diagnosis;Emphasise the impact of the KS pathogenetic trifecta encompassing HHV8 genes, KS oncogenes, and inflammatory cytokines on the present understanding of KS.

In addressing the above goals, new genomic technologies, that have not only facilitated advances in HHV8 characterisation, but have also advanced the molecular landscape of KS and detailed the intricate and complex human–virus nexuses amongst host, inflammatory, and viral proteins, are highlighted. Despite these major clinicopathological and molecular breakthroughs in KS, hiatuses exist. These gaps, therapeutic options, and treatment targets are critically reviewed.

## 2. KS in the Pre-HHV8 Era

### 2.1. The Pre-AIDS Era of Classic and Endemic KS

KS, an intermediate-grade vascular tumour, was first described by Moritz Kaposi [1], the Viennese professor of dermatology and syphilis. Prior to the 1950s, KS was recognised in its classic form exclusively [2], manifesting as multiple pigmented lesions in elderly men of Jewish and Eastern European descent [3]. In the 1960s, an endemic form of KS [2] was recognised in Uganda and the Democratic Republic of Congo (DRC), where it was reported in 9% of all malignancies in men [4].

Regarded as an enigmatic disease since the 1950s, KS has been characterised by opposing clinicopathological concepts. In their classification of KS, Taylor et al. [4] included the varying viewpoints of pathologists. Some pathologists viewed KS as a heterogeneous combination of spindle cells, vascular channels, and inflammation that was a function of a specific developmental stage. In contrast, other pathologists believed that KS was either exclusively well or poorly differentiated. Larger studies on KS in Uganda and the DRC were seminal in pioneering the first clinical classification that correlated with histopathological features [4] (Table 1).

The clinical morphological lesions that were classified as nodular, lymphadenopathic, florid, and infiltrative types correlated with a dominance of nodules and plaques, nodules, fungating exophytic, and diffusely infiltrative skin tumours, respectively [4]. The consistent presence of bone involvement and locally aggressive disease set the florid and infiltrative tumours apart from the nodular and infiltrative KS types. Microscopically, Taylor et al. [4] documented mixed cell, monocellular, and anaplastic variants. These terms portrayed the relative dominance of vasculature, spindle cells, and cellular pleomorphism within the tumours and remain poised as important microscopic diagnostic clues of KS to date.

The cluster distribution of KS in endemic areas led to the initial belief that the causes of KS were multifactorial, and included hereditary, infectious, and environmental associations. Mendelian inheritance patterns were thought not to play a role because of the lack of consanguinity and the infrequency of familial KS [5].

### 2.2. Increased Incidence of KS and AA-KS Facilitated Greater Recognition, Research, and Discussion of KS

In contrast to the 1950s and 1960s, when differing viewpoints of KS existed, the 1980s and 1990s heralded a consensus period, when multiple clinical and histopathological variations of KS became accepted, especially with respect to the size and number of neoplastic spindle cells and the accompanying inflammatory component.

In the early 1980s, KS was the most common malignancy in individuals infected with HIV [6], and these cases were thereafter classified as AIDS-associated KS (AIDS-KS). Increasing numbers of AIDS-KS were reported originating from groups of homosexual males in New York and California.

This increased incidence of KS in the 1980s and 1990s [7] led to heightened clinical awareness and histopathological recognition of KS, especially the early patch stage. Critical discussion and evaluation of the multiple forms of KS effected greater affirmation that the classification introduced by Taylor et al. [4] in 1971 was still robust and comprehensive [7,8] (Table 2). It was in New York that Chang et al. [9] discovered KSHV, which became a significant milestone in the history and diagnostic armamentarium of KS.

## 3. KS in the HHV8 Era

### 3.1. KSHV Discovery: The Diagnostic and Biological Turning Point of KS

Chang et al. [9] utilised representational difference analysis to identify and characterise the unique DNA sequences of KSHV. The observations linking HHV8 causally to KS included the following [52]:Polymerase chain reaction (PCR) detection of KSHV in all epidemiological forms of KS and in all fresh biopsies;Detection of KSHV DNA in the peripheral blood of HIV-positive individuals;The association of the incidence of classic KS and AA-KS with the prevalence of the virus in these populations and the latent expression of KSHV in tumour cells.

### 3.2. Development of the HHV8 LNA-1 Antibody: The Diagnostic Nexus

Dupin et al. [53] were the first to generate monoclonal antibodies for the recognition of HHV8-infected cells. This antibody was generated against a latent nuclear antigen encoded by open reading frame 73 (ORF73) of HHV8; this was later labelled latency-associated nuclear antigen (LANA/LNA-1). LANA is a 1162 amino acid protein consisting of multiple repeat elements. Dupin et al. [53] described the first “nuclear-stippling” pattern observed when the LANA antibody, LN53, recognised an EQEQE repeat epitope on LANA, successfully identifying the presence of HHV8-infected cells. KS may be a diagnostic challenge and pitfall when it mimics other vasoformative entities, including angiosarcoma, haemangioma, and pyogenic granuloma, amongst others (Table 1). Subsequent publications [54] demonstrated that HHV8 LNA-1 positivity was specific to KSHV infection, with 100% (50/50) of KS cases being positive and all other non-KS lesions being negative.

## 4. The Current Clinical Landscape of KS

Damania and Dittmer [55] debated whether the spectrum of KS is the same today as it was 40 years ago. A fundamental difference between these time periods is that whilst patients with late-stage AIDS are rare, KS remains the leading cancer in people living with HIV, including in the context of suppressed HIV viral loads. Furthermore, Batash et al. [56] report the occurrence of the oldest variant of KS, classic KS, in today’s populations. Classic, endemic, AIDS-associated, and iatrogenic KS subtypes are currently recognised (Table 3).

### 4.1. Classic KS

Classic KS affects 1:100,000 patients annually in Italian, Druze, Cypriot, and Greek populations [56]. The associated risk factors include previous infection with HHV8, age > 50 years, contact with volcanic soil, and bloodsucking insect bites [56]. Pelser et al. [57], however, report that volcanic soils may not be associated with KS, referencing poor exposure measures in the studies that reported this. They postulate that exposure to iron and other environmental factors is pathogenic. This notion is supported by Goedert et al. [58], who reported that this may simply have been an association by chance. Ascoli et al. [59] highlighted that insect bites may be a cofactor associated with KS after sampling and collecting insect populations in numerous areas. Two sandfly and three mosquito species whose bites promote a strong inflammatory response were identified in two dissimilar Italian rural environments where classic KS was prevalent. Males are typically affected more than females. The older literature by Guttman-Yasley et al. [60] reported a two to fifteen times greater incidence in males, whilst more recent work by Batash et al. [56] reported a two to five times greater incidence in males. Classic KS is well recognised but may still be misdiagnosed because of its mimicry of clinical entities, including arterial insufficiency and venous stasis (Table 1).

### 4.2. Endemic KS

When endemic KS was characterised by Taylor et al. [4] in the 1960s, a notable difference that set it apart from other KS categories was its aggressive clinical course. This feature of endemic KS holds true currently [61,62]. It includes bony invasion in up to 31% of patients, the presence of exophytic lesions, and aggressive tumours [61,62]. Lymphoedema is described in 17% of cutaneous manifestations [61,62]. In contrast to classic KS, endemic KS has a bimodal peak of 4 and 9 years of age and a male-to-female ratio close to 1:1 [62]. There is significant overlap in the symptomatology of the two diseases [3,61]. Endemic KS is challenged by a lack of high-quality epidemiological data because of the discovery of AIDS-KS in the 1980s [62]. Furthermore, there has been a dearth of published data on endemic or HIV-negative KS since.

Endemic KS has been differentiated from AIDS KS by El-Mallawany et al. [63], who compared cohorts of endemic and AIDS-KS in the paediatric population of Malawi. In the endemic subtype, there is a marked decrease in oral and visceral lesions [63].

### 4.3. AIDS KS

In comparison to classic KS, AIDS-KS progresses more rapidly. Anatomical involvement by AIDS-KS is characterised by greater cutaneous involvement of the trunk and lower extremities, as well as mucosal and visceral disease [63]. The latter includes lymph node and gastrointestinal tract disease. The South African ratio of males to females is approximately 2:1 [64]. The AIDS epidemic and resultant increase in AIDS-KS cases underpinned the new knowledge base of KS, including the molecular discoveries.

### 4.4. Iatrogenic KS

Israel Penn [65] was among the first to associate an increased frequency of KS following transplantation. This “post-transplant” KS is synonymous with iatrogenic KS in the modern era. In Penn’s 1979 [65] study, all patients had been subjected to various forms of immunosuppression. While local radiation [65] may be causative, corticosteroids are the most common drug associated with KS [66], and withdrawal or dose reduction in the immunosuppressant is recommended as an adequate therapeutic approach. A 2005 French and Italian study by Serraino et al. [67] found a significantly increased KS risk in transplant recipients, including kidney, heart, and liver transplants. They associated different KS patterns with progressive impairment of the patients’ immune function [67].

The pathogenesis of iatrogenic KS encompasses two proposed mechanisms: immunosuppression with reactivation or new infection. Barozzi et al. [68] demonstrated that the epithelial cells of the proximal tubules of the kidney demonstrate focal HHV-8 immunopositivity, supporting their hypothesis that these cells may serve as a reservoir for HHV8 reactivation. This was additionally supported by PCR positivity and negativity in immunopositive and immunonegative cells, respectively. Pathogenetically, new infections may be caused by HHV8-infected cells in donor organs or primary infection in the patient after sexual contact or non-sexual contact through salivary shedding.

## 5. Current Histopathological Landscape of KS

### 5.1. HHV8 and Diagnostic KS Confirmation

With the added utility of HHV8 LNA-1 as a commercially available diagnostic immunohistochemical antibody, the KS subtypes underwent rapid expansion from the late 1990s to the early 2000s. Schwartz [69] reported extensively on these advances in this time period, with 10 additional clinical morphologic variants of KS being described by 2006 [70] (Table 1 and Table 2). Traditionally, the KS clinical variants included patch, plaque, nodule (Figure 2), and lymphadenopathic types. The discovery of HHV8 and the subsequent emergence of the HHV8-IHC antibody facilitated the diagnosis of anaplastic KS and its distinction from poorly differentiated sarcoma, including angiosarcoma, undifferentiated pleomorphic carcinoma, and anaplastic lymphoma (Figure 3). Very early clinical lesions that mimic a bruise pose a histopathological diagnostic challenge. Biopsies of these early lesions may display the promontory sign (Figure 4), but this is not a specific feature of KS. The promontory sign displays dissection of collagen and a space around blood vessels (Figure 5). Eccrine glands may also be surrounded by a space. Few spindle cells may be present. The neoplastic endothelial cells that are highlighted by the HHV8 IHC stain are pivotal to the diagnosis of early KS.

### 5.2. HHV8 Immunohistochemistry: KS Subtype Expansion

The immunohistochemical HHV8 tool also broadened the clinical KS morphological list to include telangiectatic, keloidal, lymphangiectatic (Figure 5A,B), lymphangioma-like, angiomatous (Figure 5C,D), exophytic, and infiltrative variants [11]. Angiomatous KS may mimic a haemangioma. HHV8 staining, including focal staining (Figure 5B), is invaluable in distinguishing KS from a haemangioma, which is a benign vascular proliferation. By 2012, the histopathological variants included an additional 13 subtypes (Table 1) [1,2,3,7,10,11,12,54,70,71,72,73]. Novel presentations of these highlighted KS as a mimicker in a plethora of clinical lesions that would have required different therapeutic modalities. An example is KS with an abscessing morphology mimicking an abscess [15] (Figure 6).

### 5.3. HHV8 and KS: Diagnostic Traps and Pitfalls

Furthermore, HHV8 IHC is also diagnostically invaluable in navigating through potential diagnostic pitfalls and traps. Acroangiodermatitis, also known as stasis dermatitis, is characterised by the presence of oedema, variable fibrosis, granulation tissue with spindle cells and capillaries, and variable haemosiderin pigment deposition (Figure 7A,B). These features are also shared by KS. Distinguishing KS from acroangiodermatitis is critically dependent on positive HHV8 immunohistochemical staining (Figure 7B inset). Pyogenic granuloma-like KS has recently emerged as a distinct KS subtype. The raised erythematous polypoid appearance and the microscopic vascular proliferation (Figure 7C) with focal solid cellular foci may mimic pyogenic granuloma (PG). HHV8 immunonegativity (Figure 7D) underpins the diagnosis of PG and distinction from PG-like KS. PG is a reactive vascular proliferation that undergoes spontaneous regression.

### 5.4. HHV8 Immunohistochemical Staining Profiles

Whilst the HHV8-LNA1 antibody has enabled diagnostic confirmation of KS, staining may be variable [12] (Figure 8A–C) and may pose diagnostic challenges. The staining patterns of HHV8 may vary from homogenous dark nuclear staining (Figure 8A) to admixed homogenous and stippled dot-like nuclear positivity (Figure 8B) or focal, elusive dot-like HHV8-positive nuclei (Figure 8C). Patients on ARV therapy may have fibrotic nodules with only focal, scattered, positively stained cells [17]. Variable HHV8 positivity may also be observed amongst HHV8 immunohistochemistry antibody clones (personal observation, PKR). The cause of varying staining patterns (intensity and quantity) remains unknown currently.

## 6. Advances in the Molecular Front of KS

Following the discovery of KSHV by Chang et al. [9], numerous advances have been made in KS diagnostics. KSHV is a double-stranded DNA virus, belonging to the *Rhadinoviridae* genus, subfamily *Gammaherpesviriniae*. It is 170 kb long and comprises 75 genes. Whilst some genes are common to other *Herpesviruses*, the genes *K1–K15* are specific to HHV8.

The HHV8 lifecycle is biphasic and comprises sequential lytic and latent phases (Figure 9). The different phases each express unique genetic profiles, utilising different genes to regulate various viral actions required for HHV8 integration and replication [29].

### 6.1. The HHV8 Lifecycle

#### 6.1.1. Minor Impact: Lytic Phase

The lytic phase of HHV-8 is driven by a network of genes that interact to promote replication, evade immune responses, and contribute to the viral pathogenicity. Additionally, the lytic phase aids in KS tumourigenesis by a paracrine oncogenetic mechanism, where early lytic genes, such as *ORF34* and *ORF45*, continuously trigger signalling cascades, which induce transcription and translation of vascular endothelial growth factor (*VEGF*) and platelet-derived growth factor (*PDGF*). These factors act in a paracrine fashion, reactivating the same signaling cascade in the latent phase, promoting KS proliferation and angiogenesis. Key genes (Table 4) [29] aid in opposing host-induced viral apoptosis and promoting angiogenesis in KS.

*ORF50* is a crucial lytic phase gene, also known as the *replication and transcription activator* (*RTA*). *RTA* is essential for the transition from latency to lytic replication, acting as a master regulator that activates the expression of numerous lytic genes, thereby orchestrating the entire lytic cycle [74]. Following RTA, *ORF57* plays a vital role by enhancing the stability and export of viral mRNAs, ensuring efficient translation of viral proteins necessary for replication [75]. Additionally, *ORF34*, which encodes the K1 protein, is involved in the regulation of viral replication and is critical during the late stages of the lytic cycle [76].*ORF36* encodes a serine/threonine viral protein kinase and is essential for the production of new virions [77]. Additionally, *ORF45* modulates host cell responses, counteracting host immune defenses to allow for more efficient viral replication [78].*IL-6* (viral interleukin-6) mimics the host cytokine IL-6, aiding in immune evasion. This promotes cell proliferation whilst modulating immune responses [79].*vBCL-2* is a viral homolog of the cellular BCL-2 protein, which helps the virus avoid apoptosis [80]. Together, these genes facilitate the viral lifecycle and play significant roles in the oncogenic potential of HHV-8.MicroRNAs (miRNAs). Hussein et al. [37] have further identified miRNAs, miR-K12-10, and miR-K12-12, which aid in viral replication.

#### 6.1.2. Major Impact: Latent Phase

The latent phase of HHV8 expresses relatively fewer genes (Table 5) in comparison to the lytic phase. However, it is the phase that persists and allows for continuous replication. This is effected by extrachromosomal viral episomes that segregate and attach to the host cell machinery derived within daughter cells during mitosis [28,29]. The viral genome simply resides within cellular nuclei and maintains a chromatinised nuclear plasmid without further virion production [81]. Importantly, the controlled expression of a much smaller number of genes during the latent phase facilitates an important cloaking mechanism deployed by the HHV8 virus. The lytic phase is prone to early detection by the host immune system because of the higher number of genes expressed, leading to a greater expression of viral proteins. Thus, the KSHV genome attempts to maintain the latent phase of infection; re-entry into the lytic cycle is extremely controlled [28,29].

### 6.2. Human–Viral Protein Interactions

The interplay between human proteins and viral proteins from human HHV-8 is crucial in facilitating the growth and progression of KS. HHV8 infection may be facilitated by specific human proteins, including heparan sulphate, integrin, and ephrin [81]. These proteins may act as receptors for HHV8 and guide attachment of the HHV8 virion to the cell membrane.

#### 6.2.1. LANA

Encoded by *orf73*, LANA is viewed as the most impactful latency-associated protein [82]. LANA provides HHV8 with the ability to recruit host machinery to the viral genome by binding to various host loci. LANA typically binds to host promoters that are already in an open chromatin formation, and following binding, it can lead to inhibitory effects [28,29,41].

#### 6.2.2. Viral Protein Interactions

LANA can interact with the viral inhibitory protein *vFLIP* to facilitate a repressive chromatin state with the expression of a few viral genes. Nucleophosmin is a histone modifier that controls transcription and chromatin organisation. Nucleophosmin is phosphorylated by vCyclin, and this phosphorylation enables the binding of LANA [41].

#### 6.2.3. Viral Human Protein Interactions

LANA can bind and thereafter inhibit *p53* [41]. This further downregulates the activation of p53-dependent reporter genes, resulting in chromosomal instability. The p53 protein plays a central role in the regulation of apoptosis and aids in controlling the cell cycle. LANA-induced inhibition of p53 results in inhibition of apoptotic signalling pathways [36]. Whilst the LANA-p53 relationship is well reported, there are varying viewpoints on its true role over the years [30,33,34,36,40]. In 1996, Dada et al. [33] demonstrated that immunohistochemical p53 expression is increased but expressed in KS. Positive and negative cells were present in the same field, typical of the wild-type pattern. Whilst many studies globally were describing a typical p53 wild-type pattern [34], Pillay et al. [40] described a p53-associated progression path, with a higher expression associated with a more advanced clinical stage, and demonstrated this to good effect in the nodular stage.

Chudasama et al. [32] in 2015 reported that p53 signalling remains intact in primary effusion lymphoma (PEL) cells in response to p53-activating agents, thus concluding that HHV8 genes do not inhibit the p53 protein. However, this represents a major shortfall in global KS studies, especially in those conducted in developed continents, including Europe and America, where AIDS-KS is prevalent. Many of these studies conducted HHV8 research on the immortalised BC-3 cell line, which is an HHV8-infected cell line isolated from B-lymphocytes from the pleural effusion of an 85-year-old white male. Similarly, Boshoff et al. [31], in 1998, also established the BCP-1 cell line for the study of HHV8, also from a primary effusion lymphoma patient. A perpetual problem is created in the study of these immortalised cell lines, as they do not reflect the variations in both human and viral genetics under different environmental conditions. This contrast is germane when comparing the findings of Pillay et al. [40] in a high AIDS-KS population to studies of KS in other populations.

#### 6.2.4. NF-kB

The NF-kB family of signalling pathways and transcription factors was discovered by Sen and Baltimore [83] in 1986. The NFkB pathway is perpetually activated in many tumours, with a resultant upregulation of genes associated with immune cell recruitment, angiogenesis, and apoptotic resistance [84]. During the HHV8 latent phase, the HHV8 miRNA K12-1 interacts with and represses the cellular protein IkBa—the primary inhibitor of Nf-kB [85]. This results in the upregulation of NFkB and the downstream cascade of its tumourigenic pathways.

#### 6.2.5. Interleukin-6

Cellular interleukin (IL-6) is a cytokine that plays a role in acute inflammation, immune regulation, and tumourigenesis [86]. IL-6 binds to its receptor (IL-6R) and the signal transducer gp130 to produce its downstream effects in the Janus kinase (JAK)–signal transducer and activator of transcription (STAT) (JAK-STAT) and Ras-mitogen-activated protein kinase (MAPK) pathways [86]. However, whilst the pathway typically requires binding of IL-6 to the IL-6R/gp130 receptor complex, this pathway may also be triggered by a viral clone of IL-6, termed vIL-6, discovered by Moore et al. in 1996 [87]. *orfK2 * encodes for vIL-6, which displays a 25% similarity to human IL-6 (hIL-6) [88]. Structurally, vIL-6 maintains the four-helix bundle of hIL-6; however, vIL-6 may bind to gp130 directly to trigger the downstream effects on its own, without the need for IL-6R. This can further upregulate vascular endothelial growth factor (VEGF) and aid in KS tumourigenesis [88].

#### 6.2.6. B-Cell Lymphoma-2

B-cell lymphoma-2 (bcl-2) is a regulator protein controlling apoptosis, produced by the *BCL2* gene [89]. The bcl-2 family of proteins has pro-apoptotic or anti-apoptotic roles based on the varying sequence homologies of the Bcl-2 homologues [89]. The viral bcl-2 homolog (vbcl-2) produced by HHV8 is encoded by *orf16* and is typically expressed during the early lytic phase of replication [90]. It demonstrates similar functions to human bcl-2 (hbcl-2) and suppresses autophagy by interacting with beclin-1. *vBCL-2* can also interact with cellular protein Aven, which inhibits Apaf-1/caspase-9-mediated apoptosis [90].

## 7. Molecular Detection of HHV8

Pan et al. [39] have designed eight primers focused on six regions within the HHV8 genome for the detection of KSHV by PCR. Their study found that primer sets to the ORFK9 region demonstrated the highest sensitivity; the ORFK9-3 primer demonstrated 100% specificity and sensitivity on 15 HHV8-positive and 16 HHV8-negative KS cases.

One of the first methods used for quantification of the HHV8 viral genome was performed by Lallemand et al. [38] in 2000. They developed a quantitative real-time PCR assay that was highly sensitive and enabled the detection of HHV8 when measuring a minimum of 10 copies per well.

## 8. Inflammatory Drivers of KS

The development of KS has been linked to immunodeficiency, in addition to KS proliferation within an inflammatory milieu. These inflammatory cells, composed of T-cells (CD8 and CD4), B-cells, macrophages, plasma cells, and dendritic cells, are first seen in early patch and plaque stages and increase in the nodular stage [91]. This conclusion was derived in the late 1980s, with much difficulty prior to the discovery of HHV8 by Chang et al.; this biological basis of inflammation-mediated KS is undervalued in the modern era. KS was studied by means of in vitro investigations utilising cell cultures. However, maintenance of these long-term cultures was not feasible because they typically showed limited growth. In 1998, Nakamarua et al. [92] successfully maintained a long-term KS-derived cell culture by growing the KS cells with conditioned media from T-cells infected with human retroviruses. Their most successful growth was with T-cells infected with Human T-Lymphotropic Virus II, which allowed the maintenance of these cell cultures for months. Whilst they did not understand the factors induced by these T-cells, their research set the foundation for the development of long-term cell cultures of AIDS-KS cells.

In 1998, Ensoli et al. [93] reported that immune dysregulation in KS encompasses immunoactivation of CD8+ T-cells and increased levels of inflammatory cytokines with a particular dominance of Th1-type cytokines. Microscopically, they support this argument by attributing the early-stage lesions of KS (patch, plaque) to closely resembling granulation tissue. Immunohistochemical studies demonstrated a CD8+ T-cell dominance of these early patch-stage lesions, with fewer B-cells and plasma cells. The CD8+ T-cells, in association with other monocytes, produced a variety of cytokines that promoted KS proliferation.

Patients with KS may additionally present with abnormally high cytokine levels and HHV8 viraemia, prior to histopathological confirmation of KS. This may manifest as a cytokine storm composed predominantly of IL-6 and IL-10, and HHV8 viral load levels may correlate with the cytokine responses. This phenomenon was first described in 2010 by Uldrick et al. [94]. A series of six patients had severe inflammatory multicentric Castleman disease-like symptoms, and their serum viral and human IL-6 levels were significantly higher in comparison to control patients. The term KS-associated inflammatory cytokine syndrome was subsequently coined for this discovery [95].

Caro-Vegas et al. [96] described an Eastern and Central African paediatric population with KS. They concluded that paediatric KS has a different viral presentation in comparison to adult KS. A total of 34 paediatric cases were compared to 207 adult KS patients. Whilst adults in the US and African populations demonstrated undetectable HIV viral loads in one third of patients, the paediatric population did not follow this trend, and all cases demonstrated higher HIV viral loads of approximately 14,000 copies/mL, higher than adult cases. They further confirmed higher IL-6 and IL-10 cytokine levels. These cell counts were attributed to paediatric populations having a larger absolute number of circulating immune cells and T-cells in comparison to adult populations, further demonstrating the role of the inflammatory microenvironment in KS proliferation and growth.

The nuclear factor k-b (NF-kB) transcription factor plays a fundamental role in the expression of pro-inflammatory proteins within KS lesions [96,97]. Following extracellular stimulation, NF-kb translocates to the nucleus, where *NF-kB*-responsive genes are transcribed. NF-kB also aids in maintenance of latency, and numerous HHV8 viral proteins aid in activating NF-kB, including vFLIP and vGPCR. *NF-kB* is involved in canonical and non-canonical pathways. The canonical pathway [98] is primarily dysregulated in KS. This plays a role in immune and inflammatory responses. It is activated by TNF-a or IL-1 and results in T-cell differentiation and activation. Additionally, pro-inflammatory cytokines aid in dendritic cell maturation and macrophage and neutrophil recruitment [99].

## 9. Treatment of KS

The primary aim of the treatment of KS is to induce regression of the lesion or to completely excise a lesion if feasible. There is no global standardised recommendation for the treatment of KS, and many options are available. The best therapeutic options are typically tailored to the number and type of KS. The European Dermatology Forum maintains extensive guidelines for the treatment of KS. However, this may not be adopted internationally because of limitations of certain drugs and treatment modalities worldwide [48].

### 9.1. Current Treatment Options

#### 9.1.1. Antiretroviral Therapy—Anti-HIV

Initiation or maintenance of antiretroviral therapies (ARTs) is vital in HIV+ patients. In AIDS-KS, limited additional therapy is needed, with regression seen as HIV viral load decreases and CD4 counts increase. If ART and local therapies are ineffective, systemic therapy is indicated. Ngalamika et al. [50] recommend immediate imitation of ART combined with systemic chemotherapy in the nodular KS, including Adriamycin/Doxorubicin, Bleomycin, and Vincristine (ABV), as a first-line therapy regimen.

#### 9.1.2. Antiviral Therapy—Anti-HHV8

Drugs that block herpesvirus DNA synthesis were the first reported to inhibit HHV8 viral replication. The only proven drug to suppress HHV-8 replication was ganciclovir/valganciclovir [100]. This was confirmed by a reduction in the viral quantity identified in oral samples. Data from small observational studies [100] have not indicated better outcomes with the use of HHV-8 DNA synthesis inhibitors, and a larger role is seen to be played by the use of ART in treating HIV and increasing CD4 counts [101].

#### 9.1.3. Surgical Therapy

Surgical therapy is restricted to cutaneous or oral lesions that may be excised by means of wide local excision. Care should be taken to ensure that excision margins are deep enough to prevent local recurrence, and there should be histological confirmation of a complete excision.

#### 9.1.4. Chemotherapy

Localised or systemic chemotherapy routes may be employed. Local routes may include focal injections of vinblastine and sodium tridecyl sulphate. Electrochemotherapy has aided the efficacy of local chemotherapy, with ECT–bleomycin combinations demonstrating complete remission in up to 65% of treated lesions. Systemic chemotherapeutics are typically reserved for extensive, painful, infiltrative tumours with visceral involvement. Agents may include doxorubicin, paclitaxel, etoposide, and irinotecan. Paclitaxel, in combination with ART, has been demonstrated to be superior to oral etoposide and combined bleomycin and vincristine in a randomised trial performed by Krown et al. [46].

#### 9.1.5. Immunotherapy

Local immunotherapies may utilise intralesional interferons or topical imiquimod. However, these demonstrate low response rates (<35% rate of partial remission). Systemic interferons include PD-1 inhibitors, which are administered with the intention of immune reconstitution. Galanina et al. [102] reported successful treatment of KS using nivolumab and pembrolizumab. A clinical review of nine patients evinced partial and complete remission in 5/9 and 1/9 patients, respectively. The evidence affirms that PD-1 blockade may aid in controlling the HIV infection and increasing the CD4 and CD8 counts, enabling faster reconstitution of the immune system. Topical immunotherapy includes 9-cis-retinoid acid (alitretinoin gel 0.1%), which has demonstrated a 37% total or partial response rate in AIDS-KS [103]. Udhrain et al. [104] report that pegylated liposomal doxyrubicin is the treatment of choice for AIDS-KS in Western countries, demonstrating that the pegylated liposome nanoparticle as a delivery system leads to greater tumour localization of doxorubicin with improved efficacy and decreased toxicity.

#### 9.1.6. Cryotherapy

Cryotherapy is a simple treatment modality that utilises the repeated application of liquid nitrogen to freeze and kill tumour cells. This approach is limited to patch stage KS lesions that are flat and <5 mm in diameter. It is not recommended for challenging lesions in the eyelids or the tip of the nose. Kutlubay et al. [47] managed to achieve complete treatment responses without recurrence in 19/30 patients, over an average of 3.2 sessions [47].

#### 9.1.7. Radiotherapy

Radiotherapy is highly effective, with control rates as high as 90% in classic and iatrogenic KS. Radiotherapy may also be used for deep and mucosal tumours. Variable doses may be used, depending on the size and depth of the lesion, with smaller, more frequent doses preferred for larger lesions. Patch-stage, large plaques, and nodular lesions are treatable with radiotherapy.

#### 9.1.8. Laser Therapy

Older laser therapies utilised carbon dioxide and argon lasers [105]. However, modern laser therapies include pulsed dye laser (PDL) and neodymium-doped yttrium aluminium garnet (Nd:YAG) lasers [105]. PDL is indicated for macules, superficial papules, and plaques, whilst Nd:YAG is recommended for deep papules and nodules. Whilst histopathological review of PDL-treated lesions has previously identified residual tumour cells in treated areas [106], complete remission has also been observed [107,108]. Nd:YAG has demonstrated clinical improvement in approximately 80% of cases [109].

## 10. Potential Future Treatment Targets for KS

In the 1990s, polymerase inhibitors, such as ganciclovir and cidofovir, showed potent results in tissue culture with decreased HHV8 replication. However, clinical studies have demonstrated varying success. The poor outcomes of polymerase inhibitors urged the search for novel treatment targets for KS. Antivirals that could target different phases in the HHV8 viral lifecycle represented the most promising research routes. Whilst numerous studies have been conducted on HHV8 therapeutics, two characteristics of HHV8 seem the most targetable, viz., LANA protein and angiogenic inhibitors. The tumour burden associated with KS exists predominantly in the latent phase of infection. Furthermore, LANA protein is visualisable immunohistochemically. Evaluating the LANA protein as a treatment target allows one to correlate treatment outcomes directly with the protein levels seen immunohistochemically. Additionally, KS is a tumour that is characterised microscopically by a consistent proliferation of poorly formed vessels. Targeting the genes, proteins, and pathways responsible for angiogenesis appears promising.

### 10.1. Latency Phase-Associated Viral Proteins and Pathways

The expression of LANA in HHV8-infected cells represents a potential treatment target. The c-terminal DNA-binding domain (DBD) that binds the HHV8 replication origin ensures viral genome replication and segregation during mitosis. Based on this, Calderon et al. [44] identified Mubritinib as a KSHV growth inhibitor, as it inhibits LANA DNA binding. The mechanism of inhibition remains unknown to date.

### 10.2. Anti-Angiogenic Therapies

Angiogenesis is a well-recognised hallmark of cancer that provides the blood supply pivotal to tumour self-sustenance. KS, a vasoformative tumour, induces angiogenesis via the production of angiogenic factors. Vascular endothelial growth factor (*VEGF*), platelet-derived growth factor (*PDGF*), and basic fibroblast growth factor (*FGF*) are key angiogenic genes. HHV8-infected cells also induce angiogenesis by activating the hypoxia-inducible transcription factors (HIF-1, HIF-2) within endothelial cells. HIF-1 induces the activation of VEGF, PDGF, and FGF. Sivakumar et al. [51] have demonstrated increased VEGF levels in HHV8-infected cells. Thalidomide and its derivatives (pomalidomide and lenalidomide) are typified by anti-angiogenic properties through inhibition of VEGF production. Whilst thalidomide is associated with teratogenic side effects, pomalidomide and lenalidomide are better tolerated [49].

PDGF is selectively inhibited by imatinib, a c-kit/PDGF receptor inhibitor. In a phase II AIDS-KS therapeutic trial, Koon et al. [45] reported long-term imatinib-related clinical benefits in 30% of patients. Furthermore, there is no reported pharmacokinetic toxicity with ARTs.

## 11. Kaposi Sarcoma: Return to the Past: Is KS a Neoplasm or Reactive Proliferation?

The neoplastic versus pseudoneoplastic and reactive controversial theories of KS began since the discovery of KS and remain pertinent but debatable to date.

### 11.1. Clinicopathological Theories

At the height of the AIDS KS epidemic in the eighties, John Brooks [110] hypothesised that KS was a reversible hyperplasia and utilised the many peculiarities of KS to structure his arguments [110]. These included the following:

The elderly male predominance, which spawned a hormonal link;

The multicentric nature of the disease, which often followed a symmetrical glove/stocking distribution;

Classic KS would almost always regress, which was not characteristic of sarcomas.

Whilst some believed that multicentric KS began as a single lesion that metastasised to other sites, others believed that multicentric KS arose as multiple primary lesions. The opposing viewpoints of multicentric KS encouraged investigations that formed the fountainhead for advanced discussion on the clonality and molecular biology of KS. The speculations were that metastatic lesions would be monoclonal, hence multiple primary lesions would be oligoclonal. However, the results that emerged from several studies throughout the late 1900s and 2000s produced heterogeneous findings.

### 11.2. KS Disobeys the Traditional Rules of Clonality

The concept of clonality forms the central dogma of cancer diagnostics and research. Greaves et al. described the evolution of cancer from a single process of clonal expansion, wherein any cancer would originate from a single mutant cell. Furthermore, a hallmark feature of cancer is its self-sustained growth-signalling pathways. However, KS violated this rule, as it has been demonstrated that once the HHV8 driver stimulus has been removed, KS tends to regress [43].

While many studies on KS clonality have been reported [42,43,52,111], varying methods and molecular techniques deployed by various researchers restrict the optimal comparative analyses. Monoclonal, oligoclonal, and polyclonal KS profiles have been documented [42]. In 2015, Schulz and Cesarman [71] re-echoed the 1980s hypothesis of John Brooks [110], with a modern genetic perspective. They suggest that whilst KS may commence as inflammatory polyclonal reactive lesions, the HHV8-induced genetic instability may lead to cellular mutations, resulting in monoclonal proliferation and the evolution of a true malignancy. This theory is supported by the *TP53* and *KRAS* mutations seen in KS, together with the spectrum of clinical, histological, and treatment outcomes reported with KS.

In 2022, Ruiz et al. [42] conducted a systematic review on the clonality of KS, confirming critical deficits and opposing clonality results in the global literature. There is a hiatus of comparative histopathological data in these clonality studies that may confound the results. For example, Fischer et al. [35] have described significant mutations and whole chromosomal gains and losses seen within anaplastic KS samples. Clonality discussions cannot consider conventional (patch, plaque, and nodular) KS within the same realm as anaplastic KS. Anaplastic KS consistently displays far more pleomorphism and mitoses in comparison to the conventional KS, which displays *TP53* and *KRAS * mutations. The panels employed by Fischer et al. [35] identified none of these mutations in their anaplastic KS cases.

## 12. South Africa: Poised for KS Investigation and Discovery

Whilst KS was initially discovered in European populations and the AIDS association was discovered in America, Africa has been the reservoir for the endemic and epidemic forms of KS.

In South Africa, KS is the most common cancer in HIV-infected South Africans [112]. This has led to generous research on the virology, epidemiology, and histopathology of KS in the South African context [12,17,27,91,113]. Recent South African publications have described the epidemiology [114] and treatment outcomes [115] of KS. However, avenues yet to be explored in South Africa include modern molecular approaches to diagnosis. These include an evaluation of digital PCR for potential diagnosis and comparisons of the outcomes of different HHV8 clones through immunohistochemical and next-generation sequencing.

An effective roadmap to advance KS diagnostics and treatment would begin with the long-term follow-up and monitoring of patients with KS lesions. Reviewing patients regularly and charting the KS lesions to monitor their change in size and response to therapy would prove useful in the South African setting. Patients should be reviewed for ART adherence, as this would allow for evaluation of serum HHV8 viral levels and HHV8 IHC as biomarkers, to assess how this influences growth and formation of new and pre-existing KS lesions. NGS performed on biopsy samples of KS lesions in different stages (patch, plaque, nodule) would be useful to assess if there are any mutations aiding angiogenesis or KS proliferation.

## 13. Conclusions

The understanding of KS has evolved significantly since its discovery in 1872. The major milestones include the development of clinical and histological diagnostic criteria, along with the discovery of HHV8 and its association with HIV. The introduction of modern immunohistochemical and molecular technologies broadened the landscape of KS. Several histopathological variants of KS have been described with the utility of HHV8 immunohistochemistry, enabling the diagnosis of challenging biopsies. Whilst surgical excision and ART are still critical for the management of KS, a greater understanding of the KS lifecycle and inflammatory nature of KS is heralding potential therapeutic agents that may be anti-angiogenic in nature or may target various components of the latent phase of replication. Additionally, modern treatment modalities, including topical and laser therapies, may provide non-invasive treatment options with minimal side effects.

The conflicting literature on the clonality of KS poses an ongoing challenge in the investigation of KS. There is a dearth of clinical information that precludes appropriate assessment and grouping of KS cases, as the various forms of KS represent different disease states with differing inflammatory milieux. If KS is truly proven to be polyclonal across a wide range of varying clinical states, this may aid in directing future therapeutic investigations to target anti-HHV8 viral DNA mechanisms more intensely, as opposed to targeting human DNA/proteins, which may be of lower yield. Further research into the clonality of KS requires thorough documentation of clinical and histopathological phenotypes, patient demographics, immunohistochemical reporting, and consistent longitudinal next-generation sequencing studies of a range of early- to late-stage lesions. Multicentric KS at different stages of growth also represents an ideal study cohort.

## Figures and Tables

**Figure 1 ijms-26-10058-f001:**
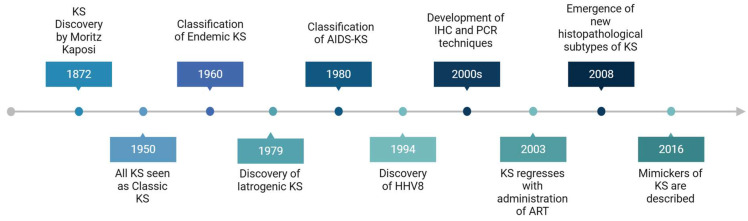
Timeline of the key discoveries in KS. Key: KS—ARTs—antiretroviral therapies, IHC—immunohistochemistry, PCR—polymerase chain reaction. Created in BioRender. https://BioRender.com/6zob4su (accessed on 11 August 2025).

**Figure 2 ijms-26-10058-f002:**
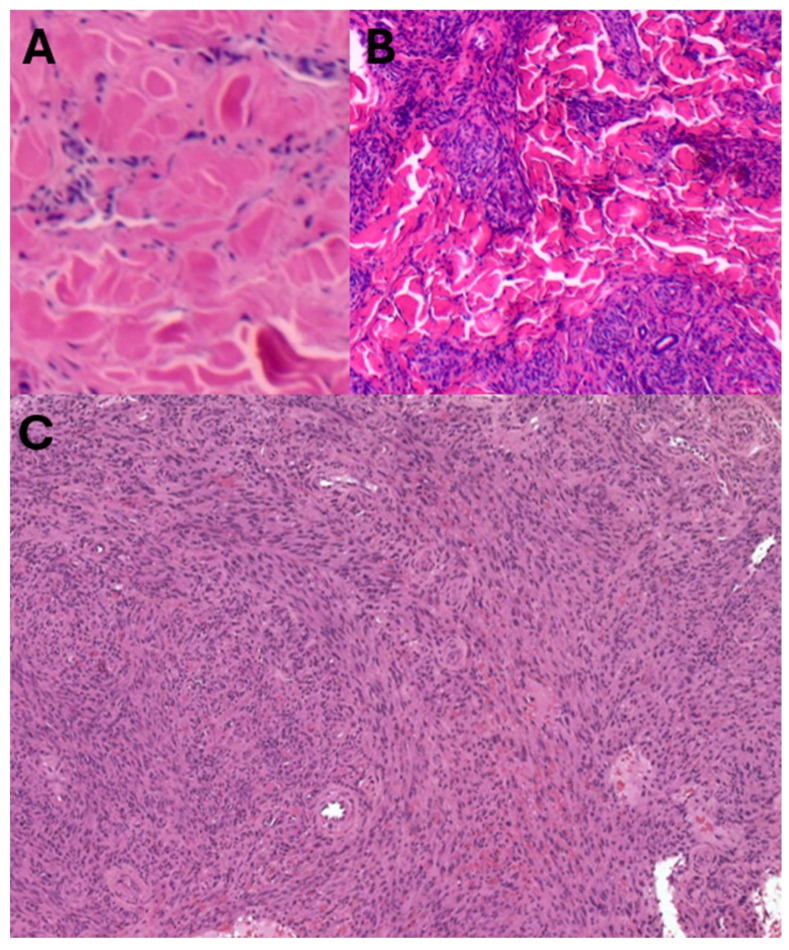
Traditional KS morphology: Sparsely cellular patch KS (**A**), increased spindle cells in plaque KS (**B**), and densely cellular spindle cell proliferation in nodular KS (**C**) [haematoxylin and eosin stain].

**Figure 3 ijms-26-10058-f003:**
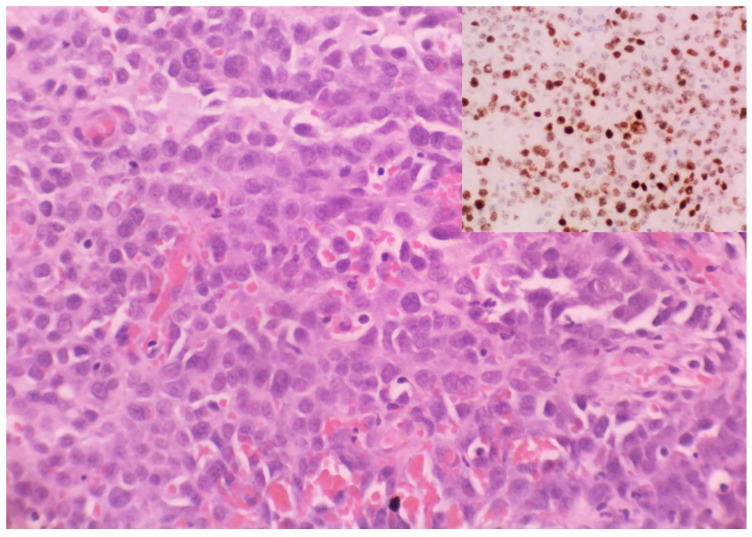
Anaplastic KS: Plump undifferentiated epithelioid cell morphology (haematoxylin and eosin). HHV8 positivity confirming KS (Inset, HHV8).

**Figure 4 ijms-26-10058-f004:**
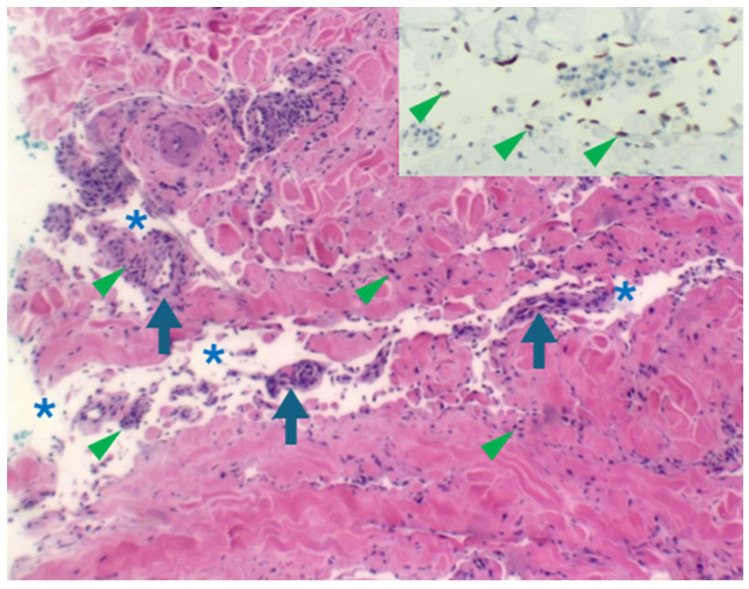
Early lesion with a patch stage morphology demonstrating the “promontory sign” (haematoxylin and eosin in which a space (asterisks) surrounds vessels (arrows) and scattered spindle cells (arrowheads). HHV8 staining of endothelial cells (inset) confirms the diagnosis of early KS.

**Figure 5 ijms-26-10058-f005:**
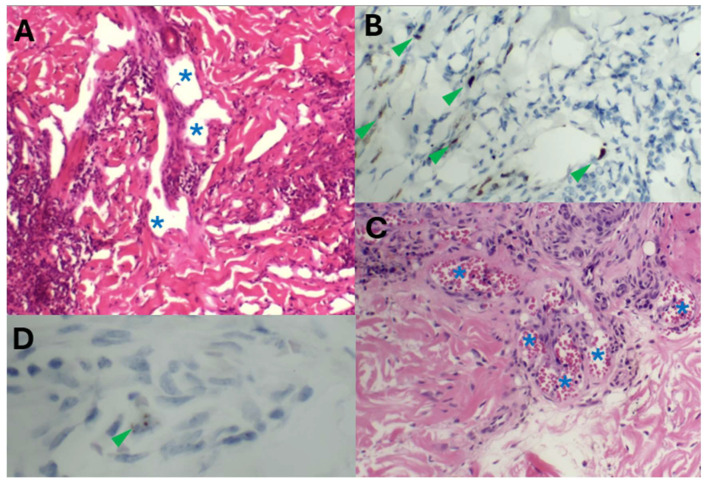
Lymphangiectatic KS (**A**), in which vascular spaces are (asterisks) devoid of erythrocytes, are lined by HHV8+ endothelial cells ((**B**), arrowheads). Angiomatous KS (**C**), in which vascular spaces with erythrocytes (asterisks) are lined by endothelial cells that show dot-like focal HHV8 immunopositivity ((**D**), arrowhead).

**Figure 6 ijms-26-10058-f006:**
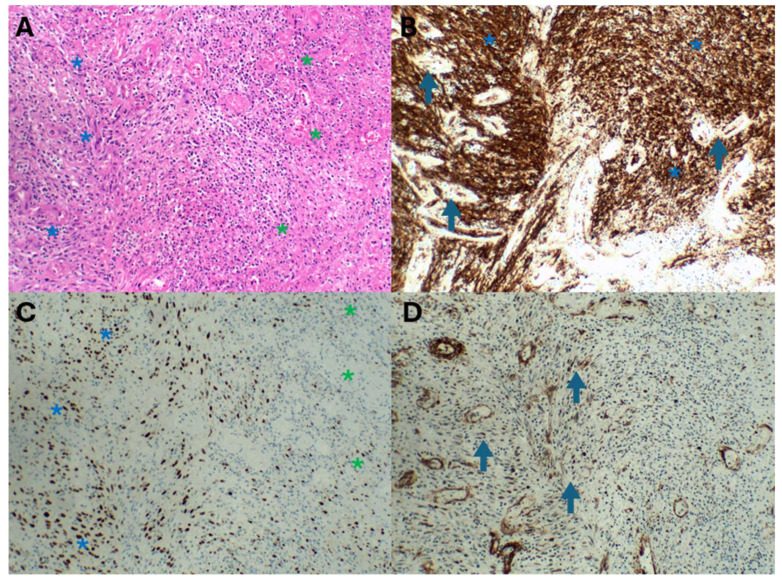
Abscessing KS ((**A**), haematoxylin and eosin) depicted by a spindle cell response (blue asterisk) and variable suppuration (green asterisk). CD34 staining (**B**) highlights the dense spindle cell KS infiltrate (blue asterisk) and vascular proliferation (arrows). HHV8 immunohistochemistry (**C**) highlights the dense positive spindle cells in KS (blue asterisks) and focal KS spindle cells (green asterisks) in the suppurative foci. Smooth muscle actin staining (**D**) confirms the myofibroblastic proliferation (arrows) and capillaries in the abscess wall.

**Figure 7 ijms-26-10058-f007:**
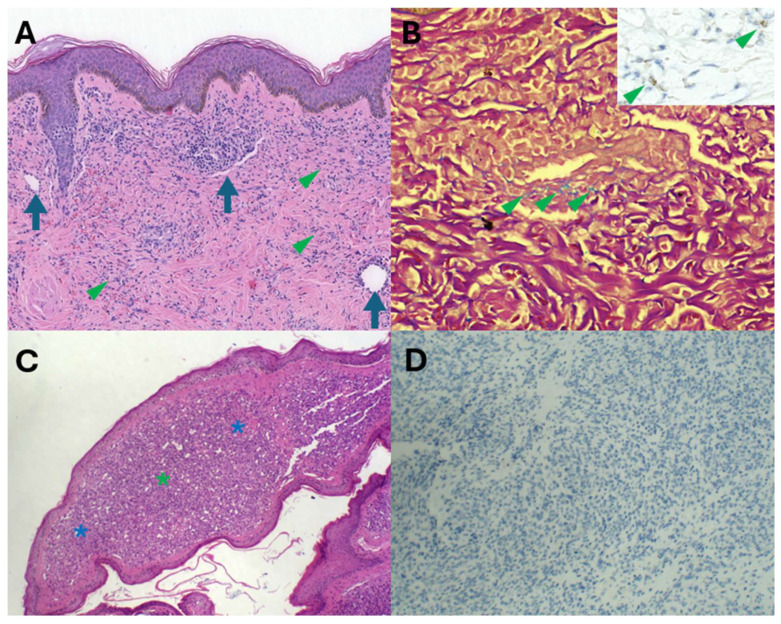
KS/acroangiodermatitis diagnostic trap (**A**,**B**): (**A**) KS and acroangiodermatitis share spindle cell (arrowheads) and vascular components (arrows) and haemosiderin pigment ((**B**), arrowheads, iron van Gieson), but HHV8 immunopositivity (inset, arrowheads) confirms KS. Pyogenic granuloma/pyogenic granuloma-like KS diagnostic trap (**C**,**D**): Polypoid vascular proliferation ((**C**), blue asterisks) and solid foci ((**C**), green asterisk) and HHV8 immunonegativity (**D**) confirming pyogenic granuloma.

**Figure 8 ijms-26-10058-f008:**
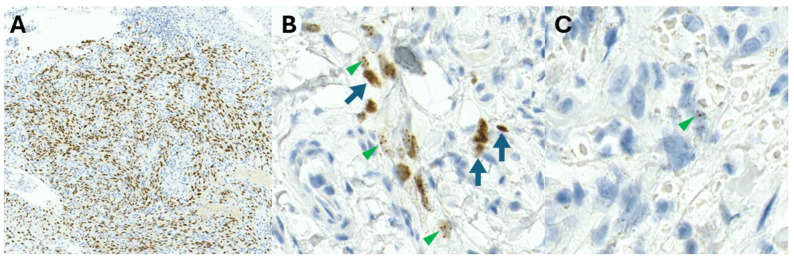
Immunohistochemical staining patterns of HHV8: (**A**) Homogenous dark nuclear staining. (**B**) Admixed homogenous (arrows) and stippled dot-like (arrowheads) nuclear positivity. (**C**) Focal, elusive, dot-like HHV8-positive nuclei (arrowhead).

**Figure 9 ijms-26-10058-f009:**
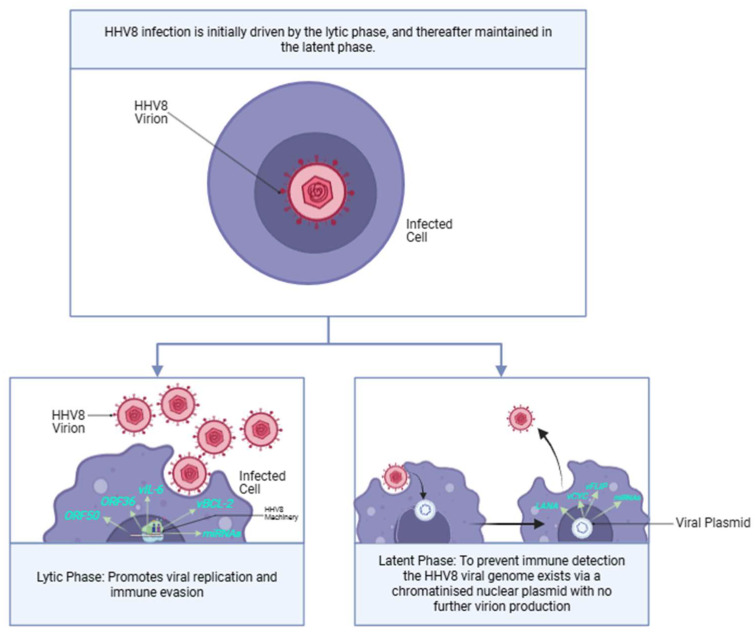
HHV8 Lifecycle and the genes implicated (blue text). Created in BioRender. https://BioRender.com/c10fp9n (accessed on 11 August 2025).

**Table 1 ijms-26-10058-t001:** Kaposi sarcoma: summary of initial correlative clinicopathological features.

Clinical Type	Main Skin Morphology	Main Microscopic Pattern	Main Microscopic Constituents
Nodular	Plaques, nodules	Mixed cell	Equal admixture of SC, VC, VS
Florid	Fungating	Monocellular Mixed cell Anaplastic	SC dominant Equal admixture of SC, VC, VS Predominant cellular pleomorphism
Infiltrative	Diffusely infiltrative	Monocellular	SC dominant Variable fibrosis
Lymphadenopathic	Nodules	Mixed cell	Equal admixture of SC, VC, VS

Key: SC: spindle cells; VC: vascular component; VS: vascular slits.

**Table 2 ijms-26-10058-t002:** Discovery of the HHV8 gene: impact on Kaposi sarcoma.

Impact	KS Feature/Profile
Expanded clinical spectrum [3,10,11]	Ecchymotic [11] Telangiectatic [11] Keloidal [11] Lymphangioma-like [3,10,11] Exophytic and infiltrative variants [3,10,11]
Widened histopathological spectrum [11,12,13,14]	Bullous [12,13] Pyogenic granuloma-like [12] Intravascular [12,14] Myoid [11,12] Glomeruloid [11,12] Micronodular [11,12]
Distinction from mimickers [15,16,17,18,19,20,21,22,23,24]	Abscessing KS from abscess wall [15] Pyogenic granuloma-like KS from PG [16] Lymphoedema [17] Anaplastic KS from pleomorphic malignant tumours [16,18] Fibroma-like KS from fibrotic nodule [16] Verrucous KS from lymphoedema [19] Lymphangiectatic KS from lymphangiectasis [17] Lymphangioma-like KS from lymphangioma [20,21] Keloidal KS from fibrotic nodule [11,22] Granulation-like KS from venous stasis [23,24]
Identification of co-lesional dual pathology [25,26,27]	KS and lymphoma [25,26] KS and TB [25,26] KS and cryptococcosis [27]
Avoidance of pitfalls [12,16,17]	Lymphoedema with or without KS [17] Radiotherapy effect [12] Drug reactions [12,16] Pseudotumoural spindle cell reactions [16]
Molecular	HHV8 genes: improved understanding [28,29,30,31,32,33,34,35,36,37,38,39,40,41]
Future directions	KS clonality and therapy [42,43,44,45,46,47,48,49,50,51]

Key: KS: Kaposi sarcoma, PG: pyogenic granuloma, TB: tuberculosis.

**Table 3 ijms-26-10058-t003:** Key features of Kaposi sarcoma subtypes.

KS Subtype	Summary of Key Features
Classic KS	Mediterranean, Middle Eastern, and Eastern European countries Mainly older men Visceral involvement is rare, at <1.4% Clinical mimicry of arterial insufficiency and venous stasis
Endemic KS	Exophytic, aggressive lesions with bony invasion in 31% of patients Lymphoedema in 17% patients with cutaneous manifestations. Bimodal peak of 4 and 9 years Male-to-female ratio close to 1:1 Decrease in oral and visceral lesions compared to AIDS-KS
AIDS KS	Progresses more rapidly than classic KS Greater cutaneous involvement of the trunk and lower extremities, as well as mucosal and visceral disease, including lymph node and gastrointestinal tract disease The South African male-to-female ratio is approximately 2:1
Iatrogenic KS	Increased frequency following transplantation, associated with immunosuppression High rates reported from Israel, Turkey, and Germany Corticosteroids are the most common drug associated with KS Withdrawal or dose reduction in immunosuppressants is recommended as a therapeutic approach May occur as a second malignancy in patients with haematolymphoid malignancies

Key: AIDS: acquired immunodeficiency syndrome; KS: Kaposi sarcoma.

**Table 4 ijms-26-10058-t004:** Key genes involved in the lytic phase of KS.

Gene	Protein	Function
* ORF50 *	RTA	Regulator of the lytic cycle Activates subsequent lytic genes
* ORF34 *	K1	Regulates viral replication
* ORF36 *	vPK (viral protein kinase)	Produces new KS virions
* IL-6 *	Viral interleukin-6	Mimics host IL-6 Promotes proliferation Facilitates immune evasion
* ORF57 *	ORF57/MTA	Contains two structurally different domains, allowing it to interact with various target RNA transcripts to act as a viral export factor to move HHV8 miRNAs out of the nucleus into the cytoplasm Acts as a viral splicing factor that promotes RNA splicing of viral pre-mRNA transcripts
* vBCL-2 *	Viral B-cell lymphoma 2	Inhibits apoptosis Supports viral survival

**Table 5 ijms-26-10058-t005:** Key genes involved in the latent phase of KS.

Gene	Protein	Function
* ORF73 *	LANA	Maintains viral genome stability and replication Binds to lytic promoters to inhibit lytic genes transcription and maintain latency
* PRF72 *	vCyc	Phosphorylates nucleophosmin (NPM) to maintain latency Acts as a functional homolog of human cyclin D2 and regulates cell cycle and proliferation
* ORF71/K13 *	vFLIP	Binds to the IkB kinase Y complex (IKKY) Activates the NF-kB pathway to facilitate HHV8-infected cell survival and proliferation Prevents apoptosis and growth arrest
* K12 *	Kaposin	Activates ERK2/MAPK pathways, enhancing the expression of cytokines and the regulation of miRNAs
* ORF50 *	RTA mRNA	Control the reactivation of HHV8 into the lytic cycle
* miRNAs *	miR-K9-5p mirK12-7 mirK12-9	Inhibit RTA expression, key to maintaining viral latency
	miR-K2 miR-k5	Angiogenesis Enhances VEGF expression

## Data Availability

No new data were created or analyzed in this study. Data sharing is not applicable to this article.

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
