# Peer review of "Recent Advances in the Histopathology, Molecular Biology, and Treatment of Kaposi Sarcoma: A Contemporary Review"

_ijms, 2025, doi:10.3390/ijms262010058_

Round 1

Reviewer 1 Report

Comments and Suggestions for Authors

In this review, Tayarv Jayd Bagratee et al. described the Histopathological, Molecular, and Treatment Advances in the oncogenesis of Kaposi’s Sarcoma. The review is well written with good Figures that explain several important aspects of Kaposi Sarcoma. Although the impact of the Latent phase is well documented, perhaps more information about the importance of the lytic phase on the paracrine oncogenesis mechanism would strengthen the review.

Minor changes:

-Page 16, line 405, ¨without the need for IL-6R¨.

Author Response

General:

The authors would like to thank the reviewer for their time and effort in reviewing this manuscript. Please see below the reviewer’s comments and corresponding author changes below. Please find changes highlighted in red in the newly uploaded manuscript.

Comment 1:
Although the impact of the Latent phase is well documented, perhaps more information about the importance of the lytic phase on the paracrine oncogenesis mechanism would strengthen the review.

Response 1:
We have revised the manuscript to include this improvement. This change can be found on Page 14, Line 318. 

Comment 2:
Minor changes: -Page 16, line 405, ¨without the need for IL-6R¨.

Response 2:
Thank you for the identification. This has been amended.

Reviewer 2 Report

Comments and Suggestions for Authors

Dear Authors,

The article “Histopathological, Molecular and Treatment Advances of Kaposi Sarcoma: A modern review” is a comprehensive and detailed research, covering both historical aspects and future perspectives.

Here are some suggestions for improving the article.

  1. The title is not clear. It would be better “Recent Advances in the Histopathology, Molecular Biology, and Treatment of Kaposi's Sarcoma: A Contemporary Review”. Because “Advances of Kaposi Sarcoma” is not correct.
  2. Since this is a review, there is no need to include the following: «The study was conducted in accordance with the Declaration of Helsinki»
  3. Since this is a review and has no Material and Methods section, it is proposed to cite the source of the figures and use the title "Typical Presentation of..." for the caption.
  4. It is desirable to highlight the recent advances in the treatment of KS and “treatment options, and opinions on clonality” in the Introduction and Conclusion section.
  5. Please, explain in the text, how the “contact with volcanic soil and bloodsucking insect bites” could be the risk factors of classical KS.
  6. There is a issue with Table 4 in Section 6.1.1 ('Minor Impact: Lytic Phase').

The text describes a specific set of genes, but the data presented in Table 4 pertains to entirely different ones. This creates a clear discrepancy between the text and the referenced table, which is misleading for the reader.

  1. Nothing is written about liposomal doxorubicin and the topical treatment, for example, “Alitretinoin” or other vitamin A derivatives.
  2. Laser therapy to destroy the blood vessels feeding the tumor is not mentioned.
  3. In conclusion the authors speculate about NGS. It is desirable to develop a roadmap or strategy, or a phased path to achieve a complex diagnostics and treatment of KS. What procedures should be done for each patient, what are the modern biochemical markers of the disease which can help the diagnosis. It is desirable to list the main of them briefly.

Author Response

Summary: 

The authors would like to thank the reviewer for their time and effort in reviewing this manuscript. Please see below the reviewer’s comments and corresponding author changes below. Please find changes highlighted in red in the newly uploaded manuscript.

Comment 1:
The title is not clear. It would be better “Recent Advances in the Histopathology, Molecular Biology, and Treatment of Kaposi's Sarcoma: A Contemporary Review”. Because “Advances of Kaposi Sarcoma” is not correct.

Response 1:
Thank you for this recommendation. We have updated the manuscript title accordingly.

Comment 2:
Since this is a review, there is no need to include the following: "The study was conducted in accordance with the Declaration of Helsinki"

Response 2:
Agree. This has been removed from the manuscript.

Comment 3:
Since this is a review and has no Material and Methods section, it is proposed to cite the source of the figures and use the title "Typical Presentation of..." for the caption.

Response 3:
Thank you for this recommendation. Please note that all figures are produced by authors Dr TJ Bagratee and Prof PK Ramdial. These images despite being original, are utilised to provide awareness to the already known histopathological landscape of Kaposi sarcoma. The wording “Typical Presentation of…” has not been used as the histopathological representation of the lesions may change on a case-by-case basis.

Comment 4:
It is desirable to highlight the recent advances in the treatment of KS and “treatment options, and opinions on clonality” in the Introduction and Conclusion section.

Response 4:
Agree. Please see a change to the introduction on Page 2, line 65. An additional change has been made to the conclusion on Page 21, line 660.

Comment 5:
Please, explain in the text, how the “contact with volcanic soil and bloodsucking insect bites” could be the risk factors of classical KS.

Response 5:
Thank you for this recommendation. We have updated the manuscript accordingly. Please see an update to the recent understanding of the controversial opinions of volcanic soil exposure as a risk factor for classical KS, along with insect bites. This may be seen on Page 6, line 157-165.

Comment 6:
There is a issue with Table 4 in Section 6.1.1 ('Minor Impact: Lytic Phase'). The text describes a specific set of genes, but the data presented in Table 4 pertains to entirely different ones. This creates a clear discrepancy between the text and the referenced table, which is misleading for the reader.

Response 6:
Thank you for the identification of this discrepancy. The paragraph has been amended, in response to both Reviewer 1 and Reviewer 2. Reviewer 1 has recommended additional review of the paracrine oncogenetic mechanism of KS, which may be seen on Page 14, line 318. This has subsequently correlated with the changes recommended by reviewer 2, and the authors have updated the table accordingly on Page 14 line 346.

Comment 7:
Nothing is written about liposomal doxorubicin and the topical treatment, for example, “Alitretinoin” or other vitamin A derivatives.

Response 7:
Thank you for recommending this update. The authors have updated the manuscript, and have described existing literature on both alitretinoin and pegylated liposomal doxorubicin. This may be seen on page 19, line 537.

Comment 8:
Laser therapy to destroy the blood vessels feeding the tumor is not mentioned.

Response 8:
Thank you for pointing this out. This has been resolved, on page 19, line 555 with a new dedicated paragraph.

Comment 9:
In conclusion the authors speculate about NGS. It is desirable to develop a roadmap or strategy, or a phased path to achieve a complex diagnostics and treatment of KS. What procedures should be done for each patient, what are the modern biochemical markers of the disease which can help the diagnosis. It is desirable to list the main of them briefly.

Response 9:
Thank you for this recommendation. This has been updated one page 21, line 656, where we build on how South Africa is uniquely poised to execute such a roadmap. This has also led to a deletion of text from page 22, line 686.

Round 2

Reviewer 2 Report

Comments and Suggestions for Authors

Dear Authors,

The paper has been fully updated, incorporating all suggestions